# The UTRs of *Leishmania donovani* vary in length and are enriched in potential regulatory structures

Franck Dumetz[1]*, Kaylee J. Watson[1], Melissa Perry[2], Robin E. Bromley[1], Anushka R. Shome[1], Julie C. Dunning Hotopp[1], Iqbal Hamza[2], David Serre[1]

**1** Institute for Genome Sciences, University of Maryland School of Medicine, Baltimore, United States of America, **2** Center for Blood Oxygen Transport and Hemostasis, University of Maryland School of Medicine, Baltimore, United States of America

* fdumetz@som.umaryland.edu

## Abstract

*Leishmania* spp. regulate gene expression posttranscriptionally, yet untranslated regions (UTRs) that can affect mRNA stability and translation remain poorly delineated. We generated a *de novo* assembled genome for *Leishmania donovani* strain 1S2D (Ld1S) using PacBio HiFi and characterized the transcriptomes of promastigotes and axenic amastigotes with Oxford Nanopore direct RNA sequencing. The genome assembly consists of 65 scaffolds totaling ~33.3 Mb. Structural comparisons to LdBPK282A1 revealed numerous rearrangements, including genes reshuffled among polycistronic transcription units and validated by RNA sequencing of polycistronic reads. Promastigote and amastigote RNA sequencing produced 469,010 and 46,729 monocistronic reads containing a spliced-leader and a polyA tail sequences, defining 8,479 transcripts and supporting 7,415 of the 7,969 annotated protein-coding genes, as well as 604 putative long non-coding RNAs. We annotated UTRs for 4,921 genes and observed that putative RNA G-quadruplexes were markedly enriched in these regions. We also noted that 31.9% and 11.5% of genes were expressed into multiple isoforms in promastigotes and amastigotes, respectively. Collectively, these data provide a genome-wide annotation of *L. donovani* genes and their UTRs and reveal widespread and stage-specific UTR length polymorphisms and, overall, points to an important role of 3' UTRs in post-transcriptional regulation in *L. donovani*.

## Author summary

*Leishmania donovani* parasites cause visceral leishmaniasis, a deadly disease affecting hundreds of thousands of people worldwide. Unlike most eukaryotes, *Leishmania* parasites do not regulate their genes at the level of transcription. Instead, gene regulation happens after the genes are transcribed, and much of this regulation likely depends on the regions of the mRNAs that are not translated

**Data availability statement:** Whole genome sequencing data are available in the NIH short read archive under the NIH BioProject PRJNA1226592 divided into BioSample SAMN47200203 (Promastigote reads) and SAMN47319287 (Amastigote reads). See Supplementary Table 7 for the details of all SRA submissions. All custom-made scripts and command lines can be found at https://github.com/Franck-Dumetz/L.donovani_ONT_G4.

**Funding:** This work was partially supported by an NIH award to the University of Maryland School of Medicine (U19 AI110820 to DS, JCDH). JCDH is supported by an MPower Professor through the University of Maryland Strategic Partnership: MPowering the State. KJW was supported by NIAID T32 AI162579. The funders had no role in study design, data collection and analysis, decision to publish, or preparation of the manuscript.

**Competing interests:** The authors have declared that no competing interests exist.

into protein, the untranslated regions (UTRs). However, UTRs in *Leishmania* remained poorly characterized. Here, we generated high-quality genome and transcriptome data for a strain of *L. donovani* commonly used in laboratory experiments. By combining state-of-the-art long-read sequencing technologies, we precisely annotated thousands of UTRs and discovered that many genes produced transcripts with variable UTR lengths. We also observed that UTRs were enriched in RNA structures called G-quadruplexes, which are known to influence gene regulation. These findings provide the most comprehensive view to date of UTRs in *Leishmania* and highlight their likely role in controlling how genes are expressed during the parasite's life cycle. Our work lays a foundation for future studies aiming at better understanding parasite biology and identifying new targets for intervention.

## Introduction

*Leishmania spp*. are the causative agents of leishmaniasis, a diverse group of diseases that affect over 12 million individuals across 98 countries, with more than 350 million people at risk of infection worldwide [1]. Leishmaniasis, a complex of diseases ranging from cutaneous to visceral leishmaniasis, presents a significant public health challenge due to the lack of effective vaccines, the limited efficacy or high toxicity of available drugs and the emergence of drug-resistant parasites [2]. Transmission occurs when infected phlebotomine sandflies inject metacyclic promastigotes into a mammalian host. Inside the mammalian host, metacyclic promastigotes are phagocytosed by professional phagocytic cells, mostly macrophages, and differentiate into intracellular amastigotes, which replicate within the acidic environment of the phagolysosome [3]. *In vitro* culture of *Leishmania* is well established for these two life stages [4,5] and both axenic promastigotes and amastigotes have proven to be good models for examining fundamental molecular and cellular mechanisms, including the regulation of gene expression [6].

In most eukaryotes, gene expression is tightly regulated at the transcriptional level by complex molecular interactions involving changes in the chromatin structure, recruitment of transcription factors, mediator proteins and RNA polymerase, and chemical modifications on the DNA sequence [7]. As a consequence, the transcription of a given gene, and its level of expression, varies according to the cell, developmental stage and environmental stimuli. By contrast, Trypanosomatids lack DNA methylation [8], do not regulate transcription initiation, and constitutively express all genes using polycistronic transcription [9]. These polycistronic RNAs, which are 100s of kb long and typically includes dozens of genes [10], are then *trans*-spliced into mature mRNAs and further edited by the addition of a spliced-leader (SL) at the 5' end and a polyadenylation (polyA) tail at the 3' end [11]. The regulation of gene expression primarily occurs posttranscriptionally and is known to involve RNA degradation [12] but, overall, remains incompletely understood.

PLOS Pathogens

In addition to differences in their transcription, Trypanosomatid genes are also more limited in their variations than in many eukaryotes. In most eukaryotes, variations in transcription start and end sites and alternative (*cis*) splicing (including exon skipping and intron retention) can lead to different isoforms expressed in different cells [13,14], and dramatically increase the diversity of transcripts generated. These processes, and their regulations, have been extensively studied in metazoans and are also present in early branching unicellular eukaryotes such as *Plasmodium* [15–17] and *Toxoplasma gondii* [18,19]. In contrast, all Trypanosomatid genes are intronless, precluding alternative splicing and reducing isoform diversity (although variations in the placement of the spliced-leader sequence have been described in *Trypanosoma brucei* [20] and transcript length isoforms have been recently described in *Leishmania infantum* [21]). This latter work on *L. infantum,* a species from the *donovani* complex, used full-length transcript sequencing to define the sites where the spliced-leader (SL) and polyA tail are added for the majority of *Leishmania infantum* protein-coding transcripts, enabling accurate annotations of the 5′ and 3′ untranslated region (UTR) boundaries and lengths [21]. This work also revealed extensive alternative trans-splicing and polyadenylation, and uncovered a rich repertoire of novel long non-coding RNAs (lncRNAs), many of which are developmentally regulated between promastigotes and axenic amastigotes [21]. In *Leishmania*, post-transcriptional regulation is largely mediated by two cis-regulatory elements, the Short Interspersed DEgenerated Retroposon (SIDER) 1 and 2 [22–24]. SIDER1 often controls stage-specificity expression [25] while SIDER2 regulates mRNA abundance [26,27]. Indeed, in *L. infantum*, comparisons of the 3′ UTRs of SIDER2-containing transcripts and transcripts without SIDER2 demonstrated their role in regulating mRNA abundance [28].

A comprehensive understanding of the unique molecular mechanisms underlying the regulation of gene expression in *Leishmania* guide the development of new drugs against leishmaniasis that do not affect the host. Unfortunately, the current paucity of genomic information about *Leishmania spp.* and other Trypanosomatids dramatically limits such studies. Few high-quality genome assemblies and annotations are available and the predicted open reading frames (ORFs) have been shown to not always be accurate [29,30]. Recently, these shortcomings have been addressed in *Trypanosoma cruzi* genomes using long-read sequencing like ONT [31] or mixed assembly of PacBio and Illumina [32], and in *T. brucei* by combining micro-C and ONT [33]. More problematic for studying posttranscriptional mechanisms, the 5′- and 3′- UTRs are typically unannotated (see, e.g., LdBPK282A1 genome on TriTrypDB): some bioinformatics algorithms have been developed to identify UTRs in *Leishmania* genes [34] but the accurate determination of the gene boundaries requires empirical data. RNA-seq data are currently available for *L. mexicana* [35], *L. major* [30] and *L. donovani* [29] but those data were generated using short-read sequencing technology, which is not ideal to accurately define transcripts, especially in organisms like *Leishmania* where polycistronic transcripts may also be sequenced and complicate the delineation of the boundaries of fully-processed mRNAs. Here, we used a combination of long-read sequencing technologies to (a) re-sequence and *de novo* assemble the genome of the 1S2D strain of *L. donovani* using Pacific Bioscience (PacBio) HiFi chemistry and (b) rigorously annotate ORFs and 5'- and 3'-UTRs of promastigote and amastigote parasites using Oxford Nanopore Technologies (ONT) Direct RNA Sequencing. We used these combined data to comprehensively define gene boundaries and isoforms in *L. donovani* promastigotes and amastigotes, and analyzed the newly defined UTRs for the presence of putative regulatory elements.

## Results

### *De novo* genome assembly of the *Leishmania donovani* MHOM/SD/62/1S strain

We generated 375,935 high-quality consensus sequences totaling 4.57 billion bases from DNA extracted from parasites of the *Leishmania donovani* MHOM/SD/62/1S strain (later referred to as Ld1S), an isolate from Sudan collected in the second half the 20th century [36]. After *de novo* assembly, we obtained 65 scaffolds accounting for 33.1 Mbp and with a N50 of 789 kbp (Table 1). For comparison, the reference genome of *Leishmania donovani* MHOM/NP/03/BPK282/0cl4, referred to as LdBPK282A1, a strain isolated in Nepal in 2003 [37], is 32.4 Mbp long and is composed of 36 scaffolds representing the complete 36 chromosomes but does not comprise a maxicircle scaffold (Table 1). Seventeen of the Ld1S

**Table 1. Summary of Ld1S assembly and annotation and comparison with LdBPK282A2.**

| | Ld1S assembly | LdBPK282A1 |
|---|---|---|
| | Genome statistics | |
| Total length (bp): | 33,315,597 | 32,444,968 |
| Fragments: | 65 | 36 |
| Maxicircle present | yes | no |
| Fragments N50 (bp): | 789,432 | – |
| Largest fragment (bp): | 2,500,085 | – |
| | Annotation statistics | |
| # of transcripts | 8,412 | 8,098 |
| # of mRNAs with both UTRs annotated | 4,921 | 0 |
| # of 5' UTRs annotated | 5,591 | 0 |
| # of 3' UTRs annotated | 6,632 | 7 |
| # of lncRNAs | 1,149 | 0 |
| # of tRNAs | 157 | 2 |
| # of rRNAs | 7 | 4 |

*de novo* assembled scaffolds generated here corresponded to entire LdBPK282A1 chromosomes (see, e.g., Fig 1A, S1 Fig), while the remaining 14 LdBPK282A1 chromosomes were each represented by two to five contigs (e.g., Fig 1B, S2 Fig). Interestingly, one scaffold of Ld1S was circular and corresponded to the sequence of the maxicircle, the Kinetoplastida equivalent of the mitochondrial genome. This sequence had not been assembled in LdBPK282A1 but is similar in its organization to the maxicircle sequences available for other *Leishmania* species [38] (S3 Fig). Note that the Ld1S maxicircle sequence is likely the full-length sequence as i) it is circular and ii) significantly longer at 29,691 bp than the previously generated maxicircle sequences (that do not include the variable region).

Comparison of the two genome sequences revealed 24,448 single nucleotide variants (S1 Table) and 2,324 sequence rearrangements (S2 Table). Out of the 2,324 rearrangements, 38 were larger than 10 kb and included 30 insertions/deletions, 2 inversions and 6 translocations (Fig 1C, S1 and S2 Fig, S2 Table). Given that gene expression in *Leishmania* occurs in polycistronic transcriptional units (PTUs), that are 100s of kb long and can include dozens of genes [10], inversions and translocations may reshuffle genes among PTUs and affect their expression. For example, the two inversions on chromosome 7 cause 14 genes (in green and yellow on Fig 1D) located in the 3rd PTU in LdBPK282A1 and transcribed from the minus strand, to be translocated to the 2nd PTU of Ld1S and transcribed from the plus strand (Fig 1D). To validate this observation, we used ONT direct RNA sequencing to sequence RNA molecules extracted from Ld1S promastigotes (see below). One 3,227 bp ONT read spanned both LdBPK_070200 and LdBPK_070070, which in BPK282 are not adjacent, confirming the rearrangement and demonstrating that it led to the expression of a different polycistronic transcript (Fig 1E). Similarly, inversions and/or translocations on chromosomes 11, 13, 18, 28 and 29 resulted in genes being located in different PTUs in Ld1S than in BPK282A1 (S4 Fig), with, in two cases, polycistronic RNA reads confirming these rearrangements (S4C and S4E Fig). Importantly, these changes of PTUs did not seem to affect, qualitatively, the expression of these genes.

## RNA transcribed from *L. donovani* promastigotes and amastigotes

We generated a total of 3,308,097 ONT direct RNA sequencing reads for promastigotes and 648,426 reads for amastigotes. Of those, 80,083 promastigote reads (2.42%) and 8,091 amastigote reads (1.25%) spanned more than one protein-coding gene (as annotated in the LdBPK282A1 genome). These reads likely correspond to polycistronic transcripts that have been incompletely processed by trans-splicing. For the rest of the analysis, we focused on reads representing RNA molecules containing no more than one protein-coding sequence.

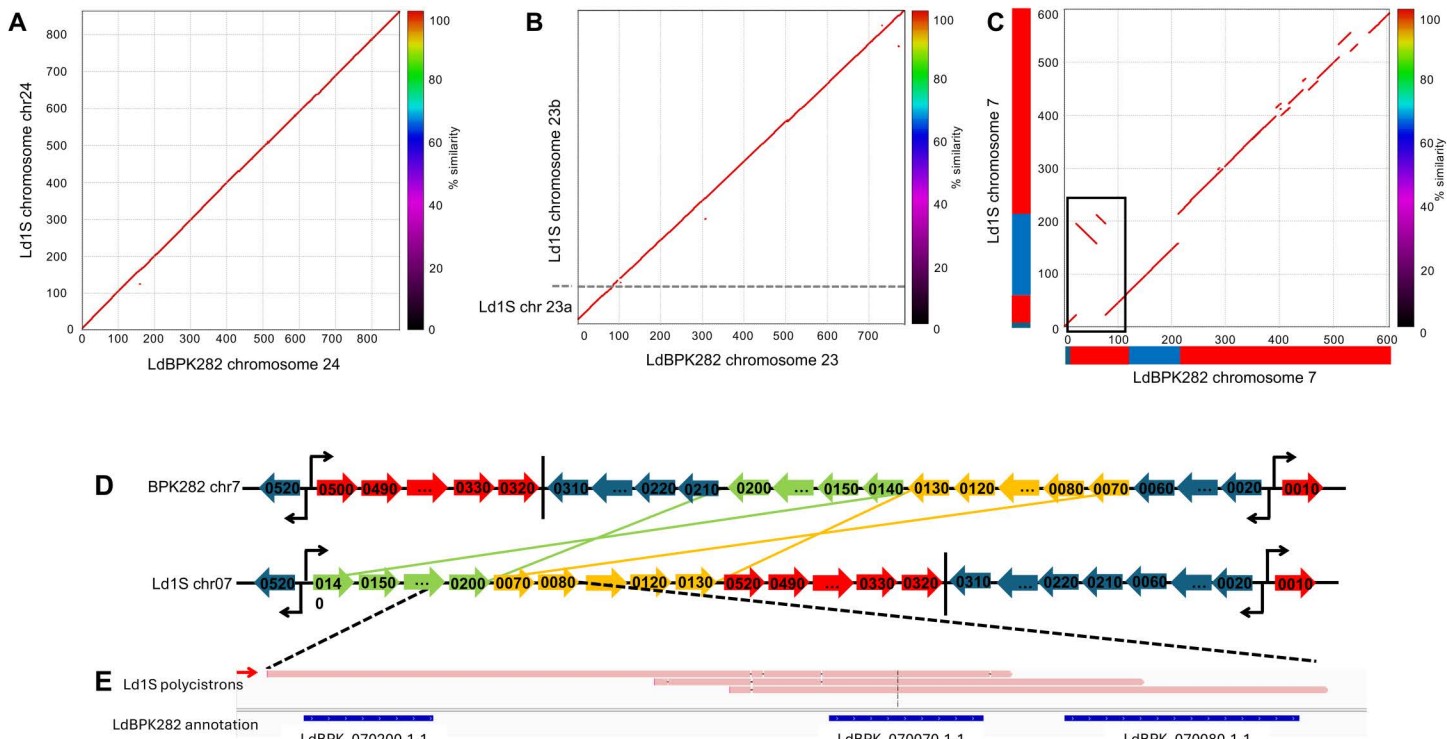

**Fig 1. Chromosome organization of Ld1S and LdBPK282A1. (A)**, **(B)** and **(C)** Display a NUCmer plot comparing the Ld1S assembly (y-axis) and the LdBPK282A1 assembly (x-axis) for chromosome 24, 7 and 23 respectively. Axis represents the size of the scaffolds in kb. Each plot represents the similarity between the two genomes (color scale of similarity on the right). Red and blue blocks represent the different Polycistronic Transcription Units (PTUs) on each chromosome: blue blocks represent PTUs on the negative strand and red blocks PTUs on the positive strand. The black box highlights the two translocations displayed in **D.** **(D)** Representation of the chromosomal rearrangement of chromosome 7 from Ld1S (at the bottom) and LdB-PK282A1 (at the top). Each arrow represents a gene; the ones in blue and red are not inverted or translocated, a vertical bar represents a transcription termination stop, and opposite arrows represent transcription switches between two PTUs. The green and yellow lines illustrate the two translocations **(D)** The panel shows polycistronic ONT direct RNA sequencing reads (in pink) spanning the rearrangement site.

To remove incompletely sequenced molecules and sequences from partially degraded mRNAs, we only considered sequences containing at least 10 As in 3'-end and a SL sequence in 5'-end (see Materials and Method for details). In total, we retrieved 469,010 reads (14%) from fully-processed mRNAs for promastigotes and 46,729 (3.33%) for amastigotes corresponding to an average coverage of 36 and 4 X, respectively. To examine whether the low proportions of reads with both a polyA tail and a spliced leader sequence were representative of a genuine biological feature of *L. donovani* RNAs or an artefact introduced bioinformatically, we performed two analyses. First, we examined the 50 first nucleotides of the sequences for any enriched motifs. The two most abundant motifs, respectively detected in 1,434,317 and 1,183,008 sequences, matched perfectly the spliced-leader sequence but only over a fraction of its length (S5 Fig), and indicated that many reads carried a spliced-leader but that these sequences failed detection due to sequencing errors and/or incomplete sequencing of the 5' end. Second, we varied the length of the spliced leader sequence queried and showed that searches with shorter sequences led to identification of more reads carrying a splice leader sequence but at the cost of a greater false positive rate (S6 Fig). These analyses indicated that the low proportions of full-length transcript sequences reflected the stringent bioinformatic cutoffs used in the analysis, rather than a biological feature of *Leishmania* mRNAs. Next, we examined whether the proportion of fully-processed mRNAs was similar across all Ld1S genes by comparing the number of raw reads observed for each gene with the number of reads with both a spliced-leader and a polyA tail (S7 Fig). Overall, the ratio of fully processed mRNA was similar for all genes, with a few exceptions mostly corresponding

to rRNAs (e.g., Ld1S.272530.1 (5.8S rRNA), Ld1S.272550.1 (18S rRNA), Ld1S.272540.1 (28S rRNA)). Histone H3 (Ld1S.100450.1) also showed a lower proportion of full-length sequences, possibly due to disproportional degradation of mRNAs but additional studies will be necessary to confirm this pattern.

In promastigotes, the 469,010 full-length RNA reads represented 8,479 transcripts of 480–43,264 bp in length (with a mean of 1,180 bp) and varying in expression from 2 to 7,482 cpm. We tested whether this variation in gene expression level was associated with the PTU a gene was located in, its relative position in the PTU or the length of the transcript, but none of these factors explained more than 8% of the variance in gene expression (S8 Fig).

Of the 8,479 promastigote transcripts, 7,415 (88.2%) were predicted to encode a protein more than 90% similar with those predicted in the LdBPK282A1 genome, three transcripts (0.036%) encoded proteins between 70% and 90% similar to those of the reference genome, and two transcripts (0.027%) encoded proteins that were not annotated in LdBPK282 genome. Finally, 1,145 transcripts (13.5%) did not encode for any protein longer than 100 amino acids and might represent long non-coding RNA (Table 1, S3 and gff file), similarly to what has been reported in *L. infantum* [21]. Note that this number of long non-coding RNAs is likely underestimated as some transcripts containing a single protein-coding sequence may actually be polycistronic and also contain a long non-coding RNA (see, e.g., S9B Fig). Overall, 7,415 out of the 7,969 (93%) protein-coding genes annotated in the LdBPK282 genome were supported by ONT monocistronic reads in Ld1S promastigotes. Note that the LdBPK282 genes not detected in the Ld1S dataset are likely due to misdetection of molecules (e.g., not sequenced until the SL) rather than missing genes (see, e.g., S9A Fig)

In Trypanosomatids, only the nuclear genome is transcribed in a polycistronic manner. The kinetoplast DNA, composed of a maxicircle (sequenced in this work) and several types of minicircles, are transcribed gene by gene. We used the direct RNA sequencing reads to annotate the unedited maxicircle genes of Ld1S. All the genes described in the *L. major*, *L. braziliensis* and *L. infantum* maxicircles [38] were identified, in the same order and on the same orientation (S10 Fig).

Direct RNA sequencing reads are generated by ligating, at the end of the polyA tail, a motor protein that pulls each RNA molecule through a nanopore. These data can therefore be used for determining the length of the polyA tail of each mRNA [39,40]. Overall, *L. donovani* mRNAs displayed a relatively short polyA tail (S11A Fig, S5 Table), with an average of 66 bp in promastigotes, significantly shorter than what has been described in mammals [40] or in *L. infantum* for example [21]. This pattern was even more pronounced for mRNAs generated from maxicircle genes (S11B Fig) that had a polyadenylation tail shorter than 50 bp (and no splice leader sequence). Further studies will be required to confirm this observation and rule out a technical artefact. Since the polyadenylation tail has been implicated in mRNA stability, we examined the correlation between the average polyadenylation tail of a gene and its expression level but failed to detect any association (S11C Fig).

### *L. donovani* UTRs are similar in length to the ones annotated in other Trypanosomatids

We used the promastigote dataset, which has the greatest sequencing depth, to annotate the UTRs of 4,921 protein-coding genes by comparing each transcript's boundaries with its predicted protein-coding sequence. The mean length of the 5' UTR was 479 nucleotides (and varied from 2 to 7,744 nucleotides), while the mean 3' UTR length was slightly larger, at 734 nucleotides (and varied from 0 to 8,267 nucleotides) (S3 Table and Fig 2A). We observed no correlation between the lengths of the 5' UTR and 3' UTR of a gene (Fig 2B) and no association with the chromosome they are located on (S12 Fig). Overall, the 5' UTR lengths of *L. donovani* were comparable to the ones of *L. major* [30], another *Leishmania* species with annotated UTRs, and were larger than the 5' UTRs of *T. brucei* (Fig 2C and S4 Table). The 3' UTR lengths were similar in all three Trypanosomatid species (Fig 2D and S4 Table). We then compared the UTR lengths of 4,770 one-to-one orthologous genes of *L. donovani* and *L. major* (Fig 2E and 2F). Interestingly, the lengths of the 3' UTRs of orthologous genes were, overall, conserved between *L. donovani* and *L. major* (although the 5' UTR of many *L. major* genes were unannotated, probably due to the difficulty of identifying full length RNA after polyA selection and Illumina short-read sequencing).

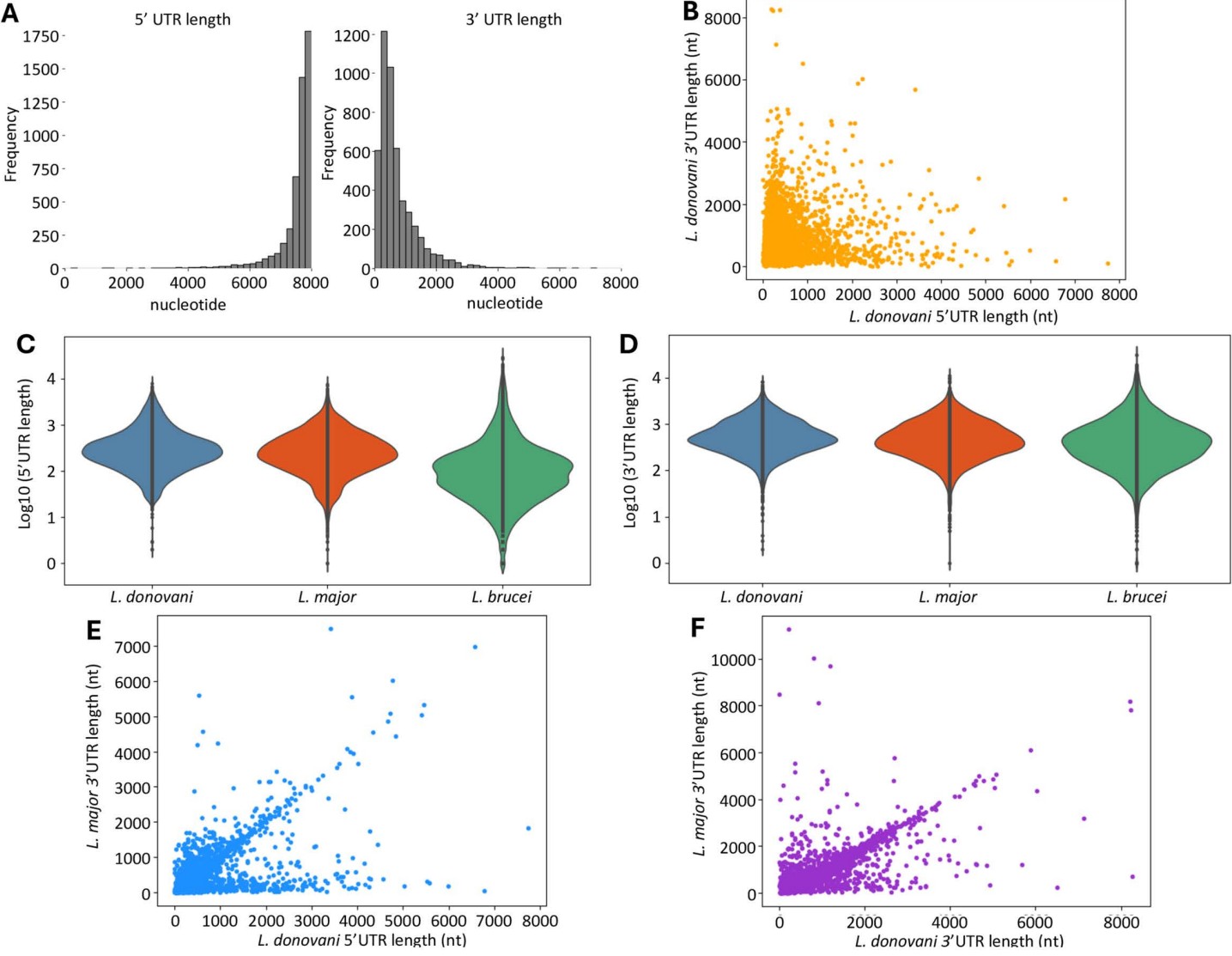

**Fig 2. Characterization of the 5' and 3' UTR lengths in *L. donovani*. (A)** Distribution of the 5' UTR (on the left) and 3' UTR (on the right) lengths. **(B)** Correlation plot between the 5' UTR length and the corresponding 3' UTR length of each transcript. Violin plots showing the distribution of the lengths (in log of bp) of the 5' **(C)** and 3' **(D)** UTRs in *L. donovani* (blue), *L. major* (red) and *T. brucei* (green). Comparison of the length of the 5' **(E)** and 3' **(F)** UTRs between *L. donovani* (x-axis) and *L. major* (y-axis). Each dot represents one gene.

### *L. donovani* UTRs often contain RNA guanine quadruplex

To preliminarily examine the possible roles of UTRs, we tested whether G-rich conserved sequences, possibly forming RNA guanine quadruplexes (rG4) (Fig 3A), were enriched in the UTRs of *L. donovani*. We screened the entire genome assembly (see Materials and Method) and identified a total of 67,993 potentially forming G4 in the genome of Ld1S. Those included 2,107 sequences located in the 5' UTRs, 5,232 in the CDS and 9,082 in the 3' UTRs (counting only occurrences on the transcribed strand). Interestingly, rG4 counts were not uniformly distributed among 5' UTRs, CDSs, and 3' UTRs ($\chi^2$ test, $p \ll 0.001$). Using the genome as a reference and after normalizing for the length of these regions (i.e., counting the number of rG4s per bp), we observed that the number of predicted rG4s in 5' UTRs roughly matched the genome

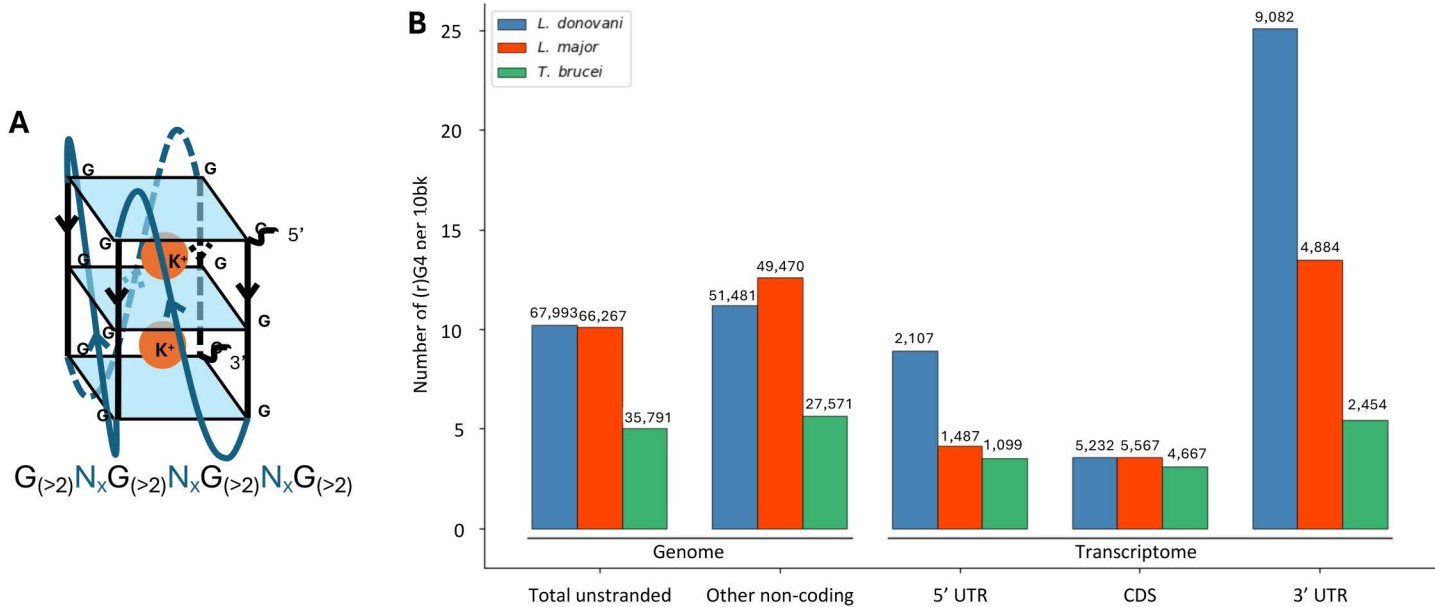

**Fig 3. rG4 are more frequent in the UTRs of *L. donovani* and *L. major* than in coding sequences. (A)** representation of a canonical rG4 and its sequence. **(B)** Number of predicted rG4 per bp (y-axis) in different genome and transcriptome regions for *L. donovani* (blue), *L. major* (red) and *T. brucei.*

average while there was a clear depletion of predicted rG4s in CDSs (OR=0.72) and a dramatic enrichment in 3' UTRs (OR=1.39)(Fig 3). One possible explanation for the lower rG4 density in CDSs is that rG4s generate secondary structures in the RNA molecules that could lead ribosome stalling and lower translation efficacy. On the other hand, rG4s were significantly enriched in 3' UTRs: out of the 6,632 3' UTRs in *L. donovani*, 3,740 (56%) contained at least one predicted rG4, with on an average of 1.3 rG4 per UTR. Interestingly, genes with predicted rG4 in their 3' UTRs were significantly enriched in protein kinase activity (Fisher's exact test, *p*-value 0.00036), catalytic activity (*p*-value 0.0048) and zinc-ion binding (*p*-value 0.04728) (S6 Table).

To further evaluate the possible role of 3' UTRs and rG4s in modulating post-transcriptional regulation in Trypanosomatids, we compared the densities of rG4 in *L. donovani, L. major* and *T. brucei.* While rG4s seemed evenly distributed throughout the *T. brucei* genome, we also observed a significant enrichment of rG4s in the 3' UTRs of *L. major* (Fig 3B), suggesting that these elements might be important for regulating gene expression in *Leishmania* species.

### *L. donovani* genes can be expressed into different isoforms with varying UTR lengths

While the previous analyses relied on the longest transcript predicted from the direct RNA sequencing data for each gene, we noticed that some genes appeared to be transcribed into multiple isoforms (see, e.g., Fig 4). We therefore screened the promastigote and amastigote datasets for transcript length isoforms, selecting transcripts supported by at least five reads or 10% of the total read counts at that locus, and differing from the previously generated annotation by at least 100 nucleotides at either the 5' or 3' UTRs (or both). Overall, out of the 4,921 genes represented by more than 10 reads (and for which multiple isoforms could be detected using our cutoffs), 1,573 genes (31.9%) were transcribed into 3,591 isoforms in promastigotes (with a mean of 2.28 isoforms per gene). Similarly, 567 genes out of all monocistrons (or 11.5%) were transcribed into 1,290 isoforms in axenic amastigotes (with a mean of 1.49 isoforms per gene). The differences in length between isoforms of the same gene were generally less than 200 bp. These modest differences are not

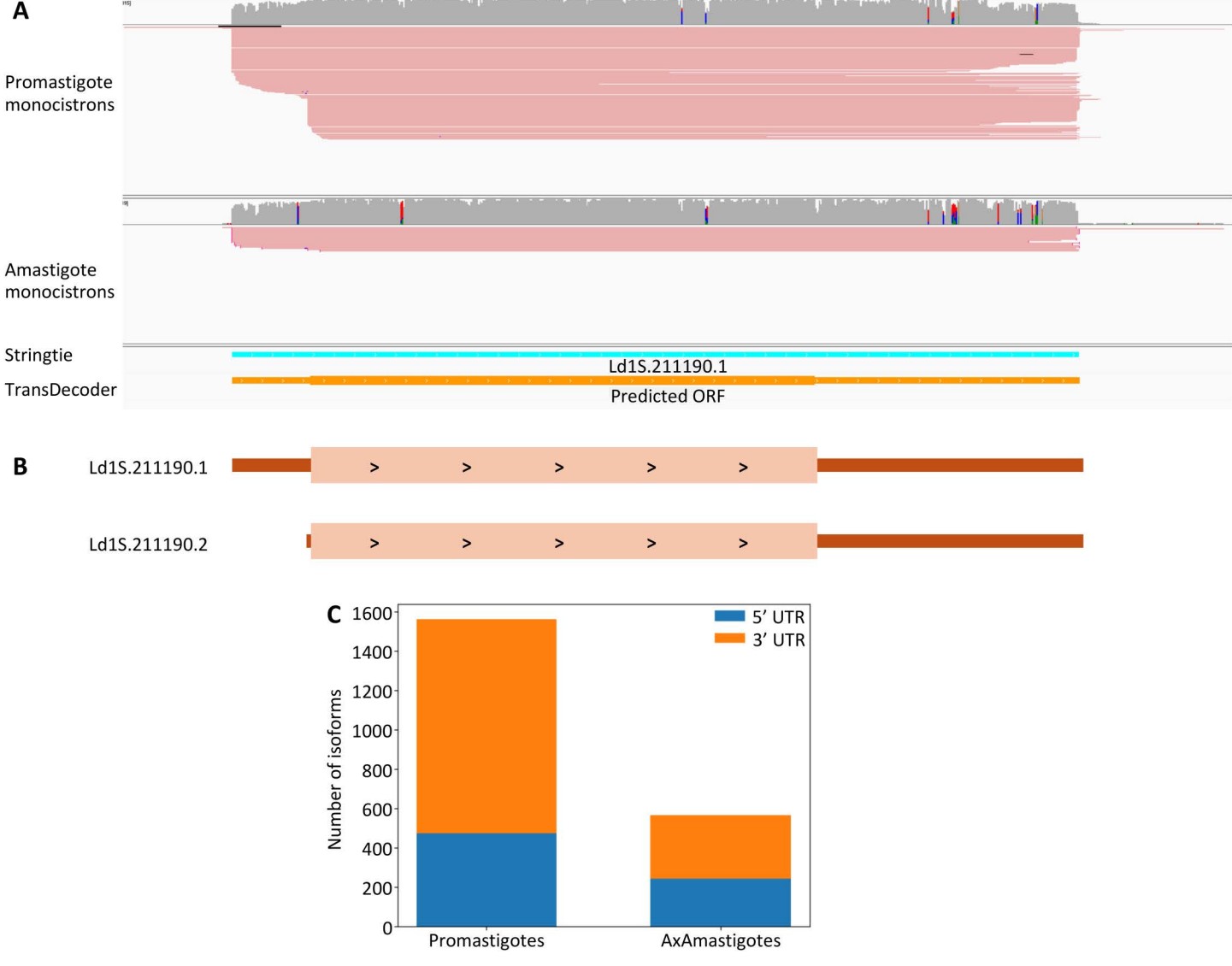

**Fig 4.** *L. donovani* **transcript length isoform.** (A) IGS screenshot showing the coverage of direct RNA sequencing reads from *Leishmania donovani* promastigotes (top panel) and amastigotes (bottom panel). The bottom panel shows the Stringtie transcript prediction as well as the TransDecoder OFR annotation. (B) Schematic representation of the two transcript isoforms: Ld1S.212290.1 is present in both promastigotes and amastigotes and Ld1S.212290.2 is only present in promastigotes.

unexpected given that intergenic regions are typically short in the *Leishmania* genome. However, these 100–200 bp differences in UTR lengths could have important functional significance since regulatory elements (e.g., rG4s or protein-binding sites) are typically very shorts and could be harbored in these variable regions.

In promastigotes, isoforms were 2.28 times more likely to involve the 3' UTR than the 5' UTR. The same trend occurred, to a lesser extent, in amastigotes (1.32-fold). This higher usage of 3' UTR length polymorphisms could reflect differences in the regulatory roles of the 3' and 5' UTRS, possibly associated with the rG4 structures described above.

## Discussion

We generated a second, high-quality, genome sequence for *L. donovani*. Comparison of the Ld1S sequence with the reference genome sequence revealed several large sequence rearrangements, including some inversions that reshuffle genes onto different PTUs. These rearrangements were confirmed by long, polycistronic, RNA reads that also demonstrated that the translocations of genes from one PTU to another had no significant impact on gene expression, consistent with the lack of transcriptional regulation in Trypanosomatids. These observations were consistent with previous reports that *Leishmania* parasites are very resilient when it comes to genome reorganization: parasites often undergo chromosomal amplification and reduction without noticeable impact on their survival and these genome rearrangements have been shown to constitute an important mechanism to adapt to the varying environments [41–43] or drug pressure [44].

The polycistronic transcription of *Leishmania* genes has hindered annotation of their UTRs: with classical short-read RNA-seq, it is challenging to differentiate reads derived from fully processed mRNAs from polycistronic transcripts. Here, we used Oxford Nanopore Technology Direct RNA Sequencing that provides full-length transcript sequences. Many of the reads generated indeed corresponded to polycistronic pre-mRNAs or incompletely processed transcripts. However, focusing on reads containing both a polyA tail and a spliced-leader sequence allowed us to identify mRNAs corresponding to 4,921 out of the 7,415 genes and robustly define their UTRs (revealing in the process many long non-coding RNAs whose roles remain unknown). Interestingly, the lengths of these UTRs, and especially the 3' UTRs, appear conserved between *L. major* and *L. donovani,* suggesting that they play important roles in the post-transcriptional regulation of gene expression. Interestingly, we observed a significant enrichment of putative RNA G-quadruplex (rG4) forming sequences, a four-stranded RNA structure formed by stacked guanine tetrads, in *Leishmania* UTRs and particularly in 3' UTRs. rG4s are key post-transcriptional regulators that have been shown to influence splicing, translation, and RNA degradation in late branching eukaryotes [45]. However, in mammals and fungi, rG4s are typically enriched in 5' UTRs [46,47] while in the only other protozoa that has being studied, *Plasmodium falciparum*, they predominantly occur in coding sequences [48]. rG4s in 3' UTRs have been found to be linked to mRNA stability [49] and RNA binding proteins (RBPs) interactions [50]. Together, given the observations of conserved 3' UTRs and the enrichment of regulatory motifs in these sequences, it is tantalizing to speculate that rG4s are one of the key mediators of post-transcriptional regulations in *Leishmania* parasites, by influencing RNA stability and/or decay, similarly to the SIDER1 and SIDER2 sequences that have been identified in the 3' UTR of ~25% of all *Leishmania* transcripts [51] and whose RNA structure is paramount to many post-transcriptional regulatory systems, including riboswitch [52]. These hypotheses will need to be tested in future studies to rigorously assess the role of these conserved UTR sequences. Importantly, the detailed catalogue of UTR sequences generated here will constitute a solid foundation for these studies and facilitate the design and/or interpretation of reporter assays, CLIP experiments, ribosome profiling or targeted UTR editing.

Finally, we observed that many *L. donovani* genes can be expressed into different isoforms, complete with polyA tail and spliced-leader sequences, primarily through variation in the length of the 3' UTRs. Given the relatively low sequencing depth of the ONT data and the very conservative criteria used in our analyses to identify multiple isoforms (e.g., a high proportion of the minor isoforms and a difference in length of at least 100 nucleotides), our estimates of the isoform diversity is likely vastly underestimated. While we sequenced "promastigote" and "amastigote" cultures, these denominations likely mask a more complex situation, and each sample probably contains a heterogeneous mixture of parasites at different stages of their development. We hypothesize that within-sample cellular heterogeneity may be partially responsible for the multiple isoforms, although this will need to be rigorously tested in future studies, for example using scRNA-seq experiments [53]. In line with the putative role of the 3' UTR in post-transcriptional regulation, these variations in 3' UTR length may be associated with the different fates of the mRNAs in different parasites. Thus, the variable UTRs at once gene could target some isoforms for rapid degradation, while a different UTR used at a different stage may stabilize the isoform and lead to robust translation of the mRNAs.

## Conclusion

We have generated a high-quality genome and a stage-specific catalogue of *L. donovani* transcripts, including their UTRs mapped at base-level resolution. Using full-length transcript sequences, we described extensive isoform diversity, mostly driven by variations of the 3′ UTR, and a strong enrichment of putative RNA G-quadruplex motifs in UTRs. Together, these findings point to the 3′ UTR as a major control hub for mRNA fate in *Leishmania*. These results will need to be followed up with various studies (*e.g.*, reporter assays, CLIP, ribosome profiling, and targeted UTR edits) to fully characterize the role of UTRs in Trypanosomatid gene regulation, facilitated by the detailed description of *L. donovani* UTRs and isoforms provided here.

## Materials and methods

### Parasite culture

Promastigotes of *Leishmania donovani* strain 1S2D (MHOM/SD/62/1S-cl2D), referred here as Ld1S, were grown at 26 °C in M199 (Earle's salts, 1× final (Gibco)) supplemented to the following final concentrations: 20% (v/v) heat-inactivated FBS (GeminiBio), 50 mM HEPES (pH 7.4) (Fisher Bioreagents), 100 µM adenine (Acros Organics), 7.6 µM hemin (Frontier Scientific), 1% penicillin-streptomycin-glutamine (Gibco), biopterin 0.0002% (w/v) (Sigma), and biotin 0.0001% (w/v) (Sigma). Hemin, biopterin, and biotin were prepared as stocks in 50% triethanolamine, 0.01 M NaOH, and 95% ethanol, respectively, and then diluted to the stated final concentrations.

Amastigotes were differentiated as previously described [54]. Briefly, log stage promastigotes were seeded at 3 x 10⁷ parasites/mL at 26°C for 24 hours in promastigote media. After 24 hours the parasites were transferred to 37°C without $CO_2$ for 24 hours. Parasites were then centrifuged at 1,500 x g for 10 minutes and resuspended in amastigote growth media (hemin 3.8 µM**,** succinate buffer pH 4.5 at 0.08 M final (from a 0.2 M succinate/0.2 M NaOH buffer used at 40% v/v (Avantor))**,** adenine 10 µM**,** glucose 0.25% (w/v)**,** trypticase 0.5% (w/v) (BD)**,** 20% (v/v) heat-inactivated FBS**,** penicillin-streptomycin-glutamine 1×, and M199 (Earle's salts, 1× final)**.** Parasites were grown at 37°C, 5% $CO_2$ for 48 hours, then split again. Amastigotes were harvested after 72 hours.

### DNA extraction, PacBio sequencing and de novo genome assembly

High molecular weight DNA (≥30 kb) was extracted from 5 x 10⁷ promastigotes using the Nanobind CBB kit (Pacific Biosciences) according to manufacturer instructions. Seven micrograms of genomic DNA sample were sheared with g-TUBE (Covaris, Woburn, MA) using the 15 kb setting. Five micrograms of sheared genomic DNA were prepared into a PacBio sequencing library using SMRTbell Express Template Prep Kit 2.0 (Pacific Biosciences) following the manufacturer's instructions. After library preparation, the library was size-selected on a BluePippin instrument (Sage Science) to remove library fragments smaller than 10 kb. The library was then bound to sequencing polymerase with Sequel II Binding kit 2.2 and sequenced with Sequel II Sequencing kit 2.0 and SMRT cell 8M on the Sequel II instrument (Pacific Biosciences) for a full 18 h movie.

The PacBio circular consensus sequence (ccs) reads were *de novo* assembled using Flye (v2.9) [55]. Quast v5.2.0 [56] and MUMmer v3.23 [57] were used to assess the quality of the assembly compared to the *Leishmania donovani* reference genome sequence LdBPK282A1 (TritrypDB [58,59], v63). Contigs and scaffolds were renamed using the MUMmer output based on the LdBPK28A1 chromosomes that they aligned to, with the exception of five contigs smaller than 5,000 bp that corresponded to repeated genes such as amastin and gp63. To identify protein-coding genes, we mapped all annotated protein-coding sequences from the reference LdBPK282A1 genome onto our *de novo* assembly using tblastn and kept coding sequence with >85% nucleotide sequence similarity over the entire gene length [60,61].

### Whole-genome alignment, SNP and rearrangement extraction

Whole-genome nucleotide alignments were generated with NUCmer (MUMmer v3.23). Alignments were then filtered with delta-filter using the options "-r -q" to retain a one-to-one local mapping between reference and query (best consistent

set per reference and per query; tolerant of rearrangements). Putative single-nucleotide polymorphisms (SNPs) were extracted with show-snps using "-C -l -r", which reports SNPs only from uniquely aligned regions (excludes ambiguous overlaps; -C), includes sequence length information (-l), and sorts by reference coordinates (-r). To summarize large-scale structural differences (e.g., changes in order/orientation such as inversions, relocations/translocations, and large indels) between the assemblies, MUMmer show-diff was run on the original unfiltered delta file.

## RNA isolation and Direct RNA Sequencing

Total RNA from *in vitro* i) Ld1S mid-log promastigotes culture and ii) 7-day differentiated axenic amastigotes culture were isolated using Trizol extraction. One microgram of total RNA was then used to prepare Oxford Nanopore Technology (ONT) Direct RNA Sequencing library using the SQK-RNA002 kit (ONT). Either SuperScript III (Invitrogen) or Induro (NEB) reverse transcriptase was used to synthesize complementary cDNA (to remove secondary RNA structures and improve sequencing). Each library was then sequenced on a R9.4 flow-cell until exhaustion. Raw fast5 files were based-called using Guppy (v6.4.2) [62] using the "rna_r9.4.1_70bps_hac.cfg" model, a minimum quality score cutoff of 7 as described previously [63] and no trimming.

## Identification of polycistronic and monocistronic reads and determination of UTRs

All ONT Direct RNA Sequencing reads were mapped to the *de novo* assembled Ld1S genome using minimap2 (v2.24) [64] using the -ont parameter. Read numbers and mapping quality were determined using SAMtools (v1.17) [65] with the "-F 2308" filter only keeping primary mapping reads.

We first removed polycistronic reads by identifying reads spanning more than one protein-coding gene using custom scripts. From the remaining reads, we retrieved reads corresponding to mature mRNAs by identifying sequences containing the last 27 nucleotides (underlined in the following sequence) of the spliced-leader (SL) specific sequence for *L. donovani*, AACTAACGCTATATAAGTATCAGTTTCTGTACTTTATTG, allowing for up to six mismatches using Fuzznuc in the EMBOSS suite of tools [66]. We used the same approach to select all the reads with a polyA tail of at least 10 consecutive As. We then used Stringtie [67] to collapse reads into transcripts using the following options "--fr -f 0.5 -c 3 -l Ld1S -p 8 -L -R" and manual curated the results to exclude artefacts, unfiltered polycistronic reads, and to retain the longest transcript when multiple predicted transcripts overlapped.

From all predicted transcripts, we determined the protein-coding sequence using TransDecoder [68] and looked for orthologs in LdBPK282A1 using blast. Any transcript with a SL and a polyA tail but that did not encode for at least 100 amino acids was annotated as a putative non-coding RNA. For all coding transcripts, we determine the 5' and 3' UTR lengths by comparing the start and end positions of each transcript with the start and end of the protein-coding sequence predicted by TransDecoder.

The annotation of the maxicircle was performed manually using BLASTn on the reads themselves and the orientation provided in previous work [38].

## Multiple isoform characterization

To identify genes that were transcribed into multiple isoforms, we looked for occurrences when a minor transcript was supported by at least five reads or 10% of the total read count, whichever is higher, and overlapped from the predominant transcript but differed in at least one of its boundaries by at least 100 bp.

## Nucleotide acid motif search

We searched for motifs, to confirm presence of the SL sequence and to investigate potential regulatory sequences, within the nucleic acid sequence using MEME (v5.5.5) [69] (parameters: -dna -mod zoops -nmotifs 5 -minw 6 -maxw 50 -objfun classic -markov_order 0). Briefly, we searched for up to 5 motifs with lengths between 6 and 50 nucleotides, compared to

scrambled sequences with the same composition as background. Motifs were searched assuming each sequence contains zero or one occurrence of the motif.

We also mapped putative RNA guanine quadruplexes (rG4) in the genome of L. donovani 1S (this study), *L. major* Friedlin2021 [30] and *T. brucei* TREU927 (TritrypDB, v67) using G4Hunter [70] website (https://bioinformatics.cruk.cam.ac.uk/G4Hunter/, March 2025) with the same parameters as in Gage and Merrick [71] (window 20 and threshold 1.5). The result from individual chromosomes were assembled into on bed file and compared with the newly generated genome annotation using BEDTools "intersect" [72]. Pairwise feature comparisons within each species were assessed with 2×2 odds ratios and two-sided Wald tests on log(OR).

### Gene ontology analysis

Functional annotation of protein sequences was performed using InterProScan v5.59-91.0 [73]. Protein sequences were provided in FASTA format, and InterProScan was executed using the parameters -goterms and -pa to retrieve Gene Ontology (GO) annotations and pathway associations. Additionally, GO term enrichment was conducted in R (v4.2.0) using the topGO package [74]. Fisher's Exact Test (weight01 algorithm) was used to identify significantly overrepresented GO terms (p-value < 0.05).

### Supporting information

**S1 Fig. Comparison of the Ld1S (y axis) and LdBPK282A1 (x axis) chromosome assemblies for chromosomes assembled in a single contigs in Ld1S. See legend of Fig 1 for details.**
(TIF)

**S2 Fig. Comparison of the Ld1S (y axis) and LdBPK282A1 (x axis) chromosome assemblies for chromosomes assembled in multiple contigs in Ld1S (separated by the dashed lines).** See legend of Fig 1 for details.
(TIF)

**S3 Fig. Comparison of the maxicircle assembly of Ld1S (x-axis) with (A)** *L. infantum*, **(B)** *L. major*, **(C)** *L. adleri*, **(D)** *L. guyanensis*, **(E)** *L. braziliensis* and **(F)** *L. tarentolae* y axis).
(TIF)

**S4 Fig. Representation of chromosomal rearrangements between Ld1S (bottom) and LdBPK282A1 (top) at chromosome 11 (A), 13 (B), 18 (C), 28 (D) and 29 (E).** Each arrow represents a gene; the ones in blue and red are not inverted or translocated, a vertical bar represents a transcription termination stop, and oppositive arrows represent transcription switches between two PTUs. Chromosomes 18 and 29 have partial or full polycistronic reads supporting the rearrangement. See legend of Fig 1 for details.
(TIF)

**S5 Fig. Most abundant MEME motif identified in the 50 first nucleotides of each ONT read compared to the *L. donovani* spliced-leader.** Motif I and II were the most abundant in all ONT reads.
(TIF)

**S6 Fig. Dot plot representing the proportion of reads with an identified SL sequence in the first 30 bp (orange) or entire ONT direct RNA sequencing reads (blue) based on the length of the SL sequence searched (x-axis) in, respectively, the promastigote- (left panel) and amastigote dataset (right panel).**
(TIF)

**S7 Fig. Comparison of the number of all reads (y axis) vs reads with a spliced leader and polyA sequences (x axis) for each promastigote gene (blue dot).** Specific genes mentioned in the main text are labeled. Two points

representing Ld1S.272530.1 (5.8S rRNA) and Ld1S.272550.1 (18S rRNA) with respectively 150,143 and 53,245 raw reads and 40 and 6 after filtering for SL and polyA tail were left out because they are off range.
(TIF)

**S8 Fig. Partitioning of gene expression variance across genomic features using fixed-effect linear modeling.** A linear model was used to quantify the proportion of variance in gene expression (log$_2$-transformed CPM) explained by gene length, position within the polycistronic transcription unit (PTU), and PTU identity (PTU ID), all modeled as fixed effects. Gene length and PTU ID account for 8.0% and 4.3% of the variance, respectively, while position in PTU contributes negligibly. The majority of variance (78.3%) remains unexplained by the model, likely reflecting additional biological and technical factors.
(TIF)

**S9 Fig. IGV screenshot showing transcript misannotations due (A) to interrupted read sequencing and (B) to non-removal of a polycistron due to the presence of a non-coding RNA.** The LdBPK282A1 CDS panel represents the CDS annotation directly transferred on the genome and the Ld1S TransDecoder output panel shows the annotation deduced from transcript evidence from direct RNA sequencing.
(TIF)

**S10 Fig. Maxicircle gene organization and read-mapping profile.** Top panel: annotated maxicircle features drawn to scale along the circular genome (linearized here). Protein-coding genes are shown as boxes, with features on the forward strand plotted above the axis (blue) and features on the reverse strand plotted below the axis (red); gene names are indicated next to their corresponding features. Middle panel: coverage across the maxicircle (grey). Bottom panel: stacked alignments of individual uniquely mapped reads across the maxicircle, colored by mapping strand (blue = forward; red = reverse). The x-axis shows maxicircle position (bp); the scale bar indicates 1 kb.
(TIF)

**S11 Fig. PolyA tail length distributions across (A) all transcripts, (B) maxicircle derived transcripts and (C) the correlation between the average polyA tail length and gene expression (in absolute read count per gene).**
(TIF)

**S12 Fig. Average of the UTR length over the transcript length of the 5' UTR and 3'UTR is similar across chromosomes.** The blue boxes represent the distribution of the ratio of the 5' UTR length and the orange boxes the one of the 3' UTR.
(TIF)

**S1 Table. NUCmer summary table of SNP differences between Ld1S and LdBPK282A1.** P1 and P2, 1-based coordinates of the variant in the reference and query sequences, respectively; SUB, the reference and query characters at that site; BUFF and DIST, distances to the previous variant in the reference and query; LEN R and LEN Q, lengths of the aligned reference and query sequences for that block; FRM, alignment orientation (1 = forward, −1 = reverse, printed as reference, query: and TAGS, the sequence IDs.
(XLSX)

**S2 Table. NUCmer summary table of DNA rearrangement between Ld1S and LBPK282A1.** Columns are: SEQ, the Ld1S sequence ID; TYPE, event class; S1/E1, 1-based start and end coordinates on SEQ; LEN 1, event size on SEQ (negative values indicate reverse orientation). Event codes: BRK (alignment break), GAP (insertion/deletion between blocks), DUP (local/tandem duplication), INV (inversion), JMP (relocation/translocation "jump" between non-syntenic blocks), and SEQ (sequence-to-sequence junction reported by the aligner). Coordinates are reported in the coordinate system of SEQ; signs reflect orientation.
(XLSX)

**S3 Table. Summary table of transferred annotation for LdBPK282A1 including newly annotated lncRNA and monocistronic transcripts.**
(XLSX)

**S4 Table. Table of isoforms found in promastigotes and amastigotes.**
(XLSX)

**S5 Table. Poly(A) tail length summary.**
(XLSX)

**S6 Table. List of gene containing an rG4 in the 3' UTR related to GO term: zinc-ion binding.**
(XLSX)

**S7 Table. Table summarizing the entire BioProject submitted to NCBI.**
(XLSX)

## Acknowledgments

The authors would like to thank Maryland Genomics for their expertise and advice at preparing DNA and for the PacBio sequencing, and Amaury Maros for coding support. We would like to thank Jain George and Suvarna Nadendla for their help with the genome and annotation submission to NCBI. We also thank our support staff for the hard work they do to allow us to only focus on science.

## Author contributions

**Conceptualization:** Franck Dumetz.

**Data curation:** Franck Dumetz.

**Formal analysis:** Kaylee J Watson.

**Funding acquisition:** Julie C Dunning Hotopp, David Serre.

**Investigation:** Franck Dumetz, Anushka R Shome.

**Methodology:** Melissa Perry, Robin E Bromley.

**Supervision:** Franck Dumetz, David Serre.

**Writing – original draft:** Franck Dumetz, David Serre.

**Writing – review & editing:** Kaylee J Watson, Melissa Perry, Robin E Bromley, Anushka R Shome, Julie C Dunning Hotopp, Iqbal Hamza.

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
