## [Decision Letter · Decision Letter 0]

27 Nov 2025

PPATHOGENS-D-25-02352

The UTRs of Leishmania donovani vary in length and are enriched in potential regulatory structures

PLOS Pathogens

Dear Dr. Dumetz,

Thank you for submitting your manuscript to PLOS Pathogens. After careful consideration, we feel that it has merit but does not fully meet PLOS Pathogens's publication criteria as it currently stands. Therefore, we invite you to submit a revised version of the manuscript that addresses the points raised during the review process.

We look forward to receiving your revised manuscript.

Kind regards,

Michael Boshart

Academic Editor

PLOS Pathogens

Dominique Soldati-Favre

Section Editor

PLOS Pathogens

Sumita Bhaduri-McIntosh

Editor-in-Chief

PLOS Pathogens

orcid.org/0000-0003-2946-9497

Michael Malim

Editor-in-Chief

PLOS Pathogens

orcid.org/0000-0002-7699-2064

**Additional Editor Comments:**

The genome sequence and transcriptome of Leishmania donovani strain 1S2D represents a significant amount of work that will be a valuable addition to LeishDB for the community. However, the reviewers agree that the manuscript in its present form is very descriptive and lacks validation of biological significance of the observations, interpretation of the correlations and testable hypotheses. Additional analyses are needed to move the work beyond a purely descriptive dataset. The authors have to expand their analyses and interpretations and demonstrate clearer biological significance, e.g. along the lines suggested by the reviewers. Figures and graphical representations of the data analyses should be improved and the text edited to make it more palatable and interesting for readers of PlosPath, beyond the narrow sequencing community. This will be a major revision, and publication will depend on whether the authors can meaningfully enhance the biological conclusions drawn from their data.

**Journal Requirements:**

2) We noticed that you used the phrase 'data not shown' in the manuscript. We do not allow these references, as the PLOS data access policy requires that all data be either published with the manuscript or made available in a publicly accessible database. Please amend the supplementary material to include the referenced data or remove the references.

3) Please amend your detailed Financial Disclosure statement. This is published with the article. It must therefore be completed in full sentences and contain the exact wording you wish to be published.

4) Kindly revise your competing statement in the online submission form to align with the journal's style guidelines: 'The authors declare that there are no competing interests.'

**Reviewers' Comments:**

Reviewer's Responses to Questions

**Part I - Summary**

Reviewer #1: In this manuscript, the authors describe the de novo assembly of the L. donovani genome using long-read sequencing technologies (PAC-Bio HiFi chemistry) and nanopore RNA seq data of two life cycle stages (promastigotes and amastigotes). This allows the precise mapping of the 5´and 3´UTRs by including only fully processed transcripts (with spliced leader and poly(A)), which is not possible with short read RNA seq methods. Most of the results are similar to available data from other Leishmania strains (isoforms with different UTR lengths), and as expected for Trypanosomatids (e.g. no correlation between PTU and expression level). One new addition of the genome is the presence of the maxicircle (mitochondrial genome) that was not present before, and the finding that RNA guanine quadruplexes are enriched in the UTRs.

This reviewer has no expertise in sequencing techniques, and cannot comment on any potential issues there. Regarding the presentation of the data, overall I’m missing a bit of interpretation, comments that go beyond the pure statistics. For example, it is nice, that now a sequence of the Maxicircle is available, but does it contain the expected genes, how conserved is it in comparison to other Leishmania strains? Otherwise, this is a nice contribution to the field. Some comments below:

Reviewer #2: Leishmania species are important model organisms and of medical interest due to their ability to infect mammals including humans. Here, the authors present the genome and transcriptome assembly of the genetically tractable L. donovani 1S2D strain using a combination of long-read DNA sequencing and direct RNA sequencing. This allows the authors to identify structural variations and provide accurate annotation of transcripts (including their UTRs) as well as identify potential non-coding RNAs. While providing a genome/transcriptome assembly for this widely used leishmania strain is an important resource for the kinetoplastid research community, there needs to be more biological insight that extends the scope of the study beyond just a descriptive resource which will require additional analyses and the inclusion of additional figures.

**Part II – Major Issues: Key Experiments Required for Acceptance**

Reviewer #1: It is nice to see alternative trans-splicing in both promastigotes and amastigotes, but do the data allow to identify genes that are differentially trans-spliced between these two life cycle stages and does it correlate with change in expression?

I do find the ratio of fully processed mRNAs with only 14% for promastigotes rather low, and I wonder, whether this is caused by the definition of the lengths of the spliced leader sequence (27 nucleotides). Nanopore usually has problems with the far 5´ends. And, in fact, the authors find shorter spliced leader sequences on a large fraction of their not-included mRNAs. Would it make sense to use a shorter fraction of the spliced leader (e.g. last 18 nucleotides instead of last 27?) to define fully processed mRNAs, to get a deeper, more robust dataset?

Line 155-157: “These analyses revealed a similar ratio of fully processed mRNA for all genes, except for 5 genes: Ld1S.272530.1 (5.8S rRNA), Ld1S.272550.1 (18S rRNA), Ld1S.272540.1 (28S rRNA), Ld1S.100450.1 (histone H3) and Ld1S.363320.1 (Ld1S_lncRNA1083).”

These are mostly not mRNA genes bur rRNA or lncRNA genes, and absence of ME (and perhaps polyA) is, at least for rRNA, expected and does not mean, these are not processed. What is interesting here is histone H3, but this is not labelled in Figure S5, so I do not even know whether the ratio is higher or lower. Can the authors comment/interpret these findings?

Most important: Please make sure to include the data into TriTrypDB!

Reviewer #2: 1. The data in supplemental Table 5 which shows the size difference between the different transcript isoforms indicates most of the isoforms do not differ drastically in size (all less than 70nt). It would be useful to plot these data as a histogram to give a visual indication as to how different in size these isoforms actually are. Presumably, isoforms with very small size differences are unlikely to have a major biological effect on expression levels, whereas larger differences may indeed change transcript stability. Furthermore, it would be helpful to know what are the transcripts which have different isoforms, and what biological processes are they involved in? A simple GO analysis of this would be useful. Are there any isoforms with large size differences between amastigotes and promastigotes? In addition, it would be nice to know what are the sequence elements which are dispensable between isoforms? This could give insight into sequences which allow increased mRNA stability or select for dominant splicing variants.

2. With the direct RNA sequencing data, the authors should be able to extract information about the presence of RNA modifications, particularly N6-methyladenosine modifications on RNA molecules. There are now several computational tools available for identifying m6A modifications from direct RNA sequencing data. Did the authors attempt to extract this information from their ONT data? As far as I am aware, no one has reported the presence of m6A in any Leishmania species. Any evidence in the ONT data that this modification exists, would be of great interest.

3. Did the authors attempt to correlate polyA tail length with mRNA expression levels? The direct RNA sequencing data should be able to provide information on polyA tail length for individual transcripts. It would be interesting to know if polyA tail length contributes to post-transcriptional control of mRNA abundance in Leishmania.

**Part III – Minor Issues: Editorial and Data Presentation Modifications**

Reviewer #1: Line 18-20: check grammar

Line 251: “annotation of”

Reviewer #2: 1. The resolution of the figures is extremely low, most of the text on the figures is barely legible.

2. Line 84-85: I think there are actually multiple high quality genome assemblies available for several trypanosomatids now, see: Rabuffo et al., 2024, Nature comms., Greif et al., 2025, Hakim et al., 2024

3. Line 150-151: the authors state that only 14% of promastigote mRNAs and 3.33% of amastigote mRNAs were considered fully processed in their direct RNA sequencing dataset; is this normal? The amount seems rather low.

4. Line 454: Figure 3 should be Figure 4

5. The accession number for the direct RNA sequencing dataset is missing in the data availability section

6. In lines 214-217 the authors mention that rG4s are enriched in the 3’UTRs of kinases and zinc ion binding proteins. I could not see any analysis of this in a figure, this would be a useful addition to figure

PLOS authors have the option to publish the peer review history of their article (what does this mean? ). If published, this will include your full peer review and any attached files.

**Do you want your identity to be public for this peer review?** For information about this choice, including consent withdrawal, please see our Privacy Policy .

Reviewer #1: No

Reviewer #2: No

**Figure resubmission:**
---

## [Decision Letter · Decision Letter 1]

28 Feb 2026

Dear Dr Dumetz,

We are pleased to inform you that your manuscript 'The UTRs of Leishmania donovani vary in length and are enriched in potential regulatory structures' has been provisionally accepted for publication in PLOS Pathogens.

Best regards,

Michael Boshart

Academic Editor

PLOS Pathogens

Dominique Soldati-Favre

Section Editor

PLOS Pathogens

Sumita Bhaduri-McIntosh

Editor-in-Chief

PLOS Pathogens

orcid.org/0000-0003-2946-9497

Michael Malim

Editor-in-Chief

PLOS Pathogens

orcid.org/0000-0002-7699-2064

Reviewer Comments (if any, and for reference):

Reviewer's Responses to Questions

**Part I - Summary**

Reviewer #1: I m happy with the manuscript now.

Reviewer #2: The authors have made a good effort to address the points which were raised and have included new analyses for some of these points. The inclusion of new analyses of polyA tail lengths is a useful addition and the manuscript looks in good shape for publication.

**Part II – Major Issues: Key Experiments Required for Acceptance**

Reviewer #1: (No Response)

Reviewer #2: None

**Part III – Minor Issues: Editorial and Data Presentation Modifications**

Reviewer #1: (No Response)

Reviewer #2: None

PLOS authors have the option to publish the peer review history of their article (what does this mean? ). If published, this will include your full peer review and any attached files.

**Do you want your identity to be public for this peer review?** For information about this choice, including consent withdrawal, please see our Privacy Policy .

Reviewer #1: No

Reviewer #2: **Yes:** James Budzak

---

## [Editor Report · Acceptance letter]

Dear Dr Dumetz,

We are delighted to inform you that your manuscript, "The UTRs of Leishmania donovani vary in length and are enriched in potential regulatory structures," has been formally accepted for publication in PLOS Pathogens.

Best regards,

Sumita Bhaduri-McIntosh

Editor-in-Chief

PLOS Pathogens

orcid.org/0000-0003-2946-9497

Michael Malim

Editor-in-Chief

PLOS Pathogens

orcid.org/0000-0002-7699-206